# Biological Control of *Acinetobacter baumannii*: In Vitro and In Vivo Activity, Limitations, and Combination Therapies

**DOI:** 10.3390/microorganisms10051052

**Published:** 2022-05-19

**Authors:** Benjamin Havenga, Brandon Reyneke, Monique Waso-Reyneke, Thando Ndlovu, Sehaam Khan, Wesaal Khan

**Affiliations:** 1Department of Microbiology, Faculty of Science, Stellenbosch University, Private Bag X1, Stellenbosch 7602, South Africa; 18339697@sun.ac.za (B.H.); breyneke@sun.ac.za (B.R.); 2Faculty of Health Sciences, University of Johannesburg, Doornfontein 2028, South Africa; 221029180@student.uj.ac.za (M.W.-R.); skhan@uj.ac.za (S.K.); 3Department of Biological Sciences, Faculty of Science, University of Botswana, Private Bag UB, Gaborone 0022, Botswana; ndlovut@ub.ac.bw

**Keywords:** *Acinetobacter baumannii*, biological control, biosurfactants, *Bdellovibrio bacteriovorus*, bacteriophages

## Abstract

The survival, proliferation, and epidemic spread of *Acinetobacter baumannii* (*A. baumannii*) in hospital settings is associated with several characteristics, including resistance to many commercially available antibiotics as well as the expression of multiple virulence mechanisms. This severely limits therapeutic options, with increased mortality and morbidity rates recorded worldwide. The World Health Organisation, thus, recognises *A. baumannii* as one of the critical pathogens that need to be prioritised for the development of new antibiotics or treatment. The current review will thus provide a brief overview of the antibiotic resistance and virulence mechanisms associated with *A. baumannii*’*s* “persist and resist strategy”. Thereafter, the potential of biological control agents including secondary metabolites such as biosurfactants [lipopeptides (surfactin and serrawettin) and glycolipids (rhamnolipid)] as well as predatory bacteria (*Bdellovibrio bacteriovorus*) and bacteriophages to directly target *A. baumannii*, will be discussed in terms of their in vitro and in vivo activity. In addition, limitations and corresponding mitigations strategies will be outlined, including curtailing resistance development using combination therapies, product stabilisation, and large-scale (up-scaling) production.

## 1. Introduction

*Acinetobacter baumannii* (*A. baumannii*) is one of the primary microorganisms linked to hospital-acquired infections such as central line-associated bacteraemia, ventilator-associated pneumonia (VAP), as well as meningitis, bioprosthetic tricuspid valve endocarditis, and urinary tract infections (UTIs) [1]. The global estimated incidence rate of *A. baumannii* infections is approximately 1 million cases annually, with crude mortality rates ranging from 20 to 80% [2,3]. Previous studies have indicated that several risk factors predispose patients to *A. baumannii* infection including age (premature babies), immunosuppression, prior hospitalisation [exposure to intensive care unit (ICU)], hospitalisation duration, surgery (invasive procedures), presence of medical indwelling devices (intravascular catheters, urinary catheter, or drainage tubes), and prior or inappropriate antimicrobial therapy [4].

Moreover, the extensive resistome of *A. baumannii* hampers the efficacy of mono-therapeutic options, and while antibiotic combination therapies have been shown to exhibit in vitro and in vivo activity against various antibiotic-resistant strains, clinical trials have not provided sufficient data to confirm that combination therapies are superior for the treatment of *A. baumannii* infections [5]. In addition, *A. baumannii*’s virulome, including cellular envelope factors, outer membrane proteins, secretion systems, phospholipases, as well as biofilm formation, allows it to persist under unfavourable environmental conditions for extended time periods, enhancing the colonisation and subsequent infection of susceptible hosts [6].

There is thus an urgent need for the research and development of alternative or novel approaches that could be used for the treatment of *A. baumannii*-associated infections, with biological control therapeutic options, defined as the elimination or eradication of a population of microorganisms through the introduction of an antagonistic (predatory) microorganism or its associated secondary metabolites, garnering increased interest [7]. For example, microbially derived secondary metabolites such as biosurfactants have been described as alternative or novel antimicrobials due to their functional properties. Lipopeptides and glycolipids are of extreme interest to the pharmaceutical and medical industry as various classes exhibit broad-spectrum in vitro and in vivo antimicrobial, antibiofilm, antiadhesive activity, and low cytotoxicity [8,9,10]. While the biosurfactants exhibit promising functional properties as biological control agents against *A. baumannii*, the application thereof remains limited due to the potential development of resistance and the high cost associated with commercialisation or large-scale (up-scaling) production [11,12,13]. In addition, biological control agents including predatory bacteria [*Bdellovibrio bacteriovorus* (*B. bacteriovorus*)] and bacteriophages have been investigated as alternative or novel antimicrobials as these approaches are considered self-sustaining, highly specific, and result in low resistance frequencies, highlighting their potential use against *A. baumannii* [14,15]. However, while the biological control agents have been observed to exhibit in vitro and in vivo antimicrobial, and antibiofilm activity with limited cytotoxicity (or deleterious effects) following treatment, the potential of developing resistance and environmental stability, are major limitations impeding their potential application against *A. baumannii.*

The current review will thus provide a brief overview of *A. baumannii*’*s* environmental persistence (bacterial survival under unfavourable environmental conditions), and antibiotic resistance strategies, primarily facilitated through virulence factors and antibiotic resistance mechanisms [16]. In addition, the therapeutic potential of microbial secondary metabolites [biosurfactants (lipopeptides and glycolipids)] as well as biological control agents including predatory bacteria (*B. bacteriovorus*) and bacteriophages will be discussed in terms of their in vitro and in vivo activity, limitations such as the potential development of resistance, product stabilisation and large-scale (up-scaling) production. Correspondingly, potential mitigation strategies will focus on the methods to curtail resistance development during treatment (combination therapy with commercial antibiotics), product (*B. bacteriovorus* and bacteriophage-derived enzymes) stabilisation for application in the medical/pharmaceutical industries, and large-scale production and optimisation of the biological control agents or their derived products.

## 2. *Acinetobacter baumannii* Resistome—Antibiotic Resistance Mechanisms

*Acinetobacter baumannii* exhibits intrinsic resistance to numerous first-line antibiotics (ampicillin, amoxicillin, amoxicillin-sulbactam, aztreonam, ertapenem, trimethoprim, chloramphenicol, and fosfomycin) and can accumulate and upregulate antibiotic resistance genes through horizontal gene transfer and insertion sequences (ISs) [17]. Multidrug-resistant (MDR) *A. baumannii* strains are classified as non-susceptible to at least one agent in three or more antimicrobial classes (antipseudomonal carbapenems, antipseudomonal penicillins + beta-(β)-lactamase inhibitors, penicillins + β-lactamase inhibitors, aminoglycosides, antipseudomonal fluoroquinolones, extended-spectrum cephalosporins, folate pathway inhibitors, tetracyclines, and polymyxins); extensively drug-resistant (XDR) strains are classified as non-susceptible to at least one agent in all but two or fewer antimicrobial categories (inhibitors of cell wall synthesis, protein synthesis, and DNA or RNA synthesis), while pandrug-resistant (PDR) *A. baumannii* strains are classified as non-susceptible to any agent in all antimicrobial categories [18].

These antibiotic-resistant *A. baumannii* strains employ an extensive range of enzymatic and non-enzymatic resistance mechanisms (Table 1). For example, β-lactam resistance amongst *A. baumannii* is primarily mediated by β-lactamases, of which all four Ambler classes (A, B, C, and D) have been detected in various strains [19] (Table 1). Inherent to all *A. baumannii* isolates is the Ambler class D oxacillinase-51-like (OXA-51) enzyme, which has been observed to facilitate penicillin (benzylpenicillin, ampicillin, ticarcillin, and piperacillin) and carbapenem (imipenem and meropenem) resistance [1]. Carbapenem resistance (imipenem, meropenem, and doripenem) has also been increasing, with a resistance rate ranging from 54.7 to 64.0% recorded amongst *A. baumannii* strains [20]. Moreover, 400 different oxacillinase enzymes, which are clustered into six subgroups (OXA-23, OXA-24, OXA-40, OXA-58, OXA-143, and OXA-235), have been identified in *A. baumannii* [19] (Table 1). Non-enzymatic β-lactam resistance mechanisms have also been detected in *A. baumannii*, including the outer membrane proteins [OMPs: carbapenem susceptible porin (CarO), OmpA and Omp 33 to 36 kDa (Omp33–36)] and resistance-nodulation-division (RND) superfamily type efflux system [*Acinetobacter* drug efflux ATP-binding cassette (AdeABC)] [21] (Table 1).

In addition to β-lactamases, *A. baumannii* mediates aminoglycoside resistance through the production of aminoglycoside modifying enzymes (AMEs) including acetyltransferases (*aac(3′)-Ia* and *aac(3′)-IIa*), nucleotidyltransferases (*ant(2′)-Ia*), and phosphotransferases (*aph(3′)-Via*) resulting in resistance to tobramycin, kanamycin, amikacin, and gentamicin [22]. Accordingly, *A. baumannii* exhibits resistance rates ranging from 80.0 to 90.0% against tobramycin, amikacin, and gentamicin [23]. Broad-spectrum aminoglycoside (amikacin, gentamicin, kanamycin, and tobramycin) resistance has then been attributed to ribosomal modifications (16S rRNA methylases: *rmtA* to *rmtD*) and non-enzymatic mechanisms such as the overexpression of RND (AdeABC) or multiple antibiotic and toxin extrusion (MATE: AbeM) superfamily efflux pumps [24,25] (Table 1). Similarly, efflux systems including the MATE superfamily efflux pump (AbeM), RND superfamily type efflux systems (AdeABC, AdeFGH, and AdeIJK), and Small Multidrug Resistance (SMR) protein family (AbeS) have been associated with quinolone resistance amongst *A. baumannii* strains [26]. Tetracycline (doxycycline and minocycline) and glycylcycline (tigecycline) resistance (ranging from 0 to 61.7%) has also been associated with two efflux pump systems including, the RND superfamily type efflux systems (AdeABC and AdeIJK), Major Facilitator Superfamily (MFS: TetA and TetB), and ribosomal protection proteins [Tet(M), Tet(W), Tet(O), and Tet(S)] [27,28] (Table 1).

Apart from the efflux systems, quinolone (ciprofloxacin and levofloxacin) resistance, ranging from 75.0 to 97.7%, has been found to be associated with mutations of the DNA gyrase (*gyrA* and *gyrB*) and topoisomerase IV (*parC*), and plasmid-mediated quinolone resistance genes (*qnrA, qnrB,* and *qnrS*) [29,30] (Table 1). Resistance to tetracycline and glycylcycline (tigecycline) may also be plasmid mediated, with recent studies detecting *tet*(X3), *tet*(X4), *tet*(X5), and *tet*(X6) genes amongst non-*Enterobacteriaceae* including *A. baumannii* [31,32].

Subsequently, the global emergence of MDR, XDR, and PDR *A. baumannii* isolates has led to a resurgence in the use of the last-resort antibiotic, colistin (polymyxin E). However, numerous studies have reported on varying degrees of colistin resistance and heteroresistance amongst *A. baumannii* [33].

Chromosomally encoded colistin resistance mechanisms amongst *A. baumannii* strains have primarily been associated with: (1) lipopolysaccharide (LPS) modification (*pmrA* and *pmrB* gene mutations; IS*Aba1* insertion upstream of the PmrC homolog EptA (*eptA*) and NaxD); (2) LPS loss (*lpxA*, *lpxC*, and *lpxD* gene mutations; or IS*Aba11* in *lpxA* or *lpxC* genes); the (3) downregulation of export and/or stabilisation proteins of the outer membrane precursors (LpsB, LptD, VacJ, and PldA); and the (4) reduction in cofactor gene expression (biotin) (extensively reviewed by Lima et al. [34]) (Table 1). More recently, a novel plasmid mediated mobile colistin resistance (*mcr*) gene, previously conserved amongst *Enterobacteriaceae*, was detected in *A. baumannii* strains [35]. The *mcr* gene encodes for a phosphoethanolamine (PEA) transferase that transfers the PEA to lipid A, resulting in a more cationic LPS and thus the repulsion of colistin [34]. To date, two variants, namely, *mcr-1* and *mcr-4.3*, have been detected in various strains of *A. baumannii*; however, no direct correlation was made by Hameed et al. [36] and Ma et al. [37] between the presence of the genes and colistin resistance. Martins-Sorenson et al. [38], however, reported on a direct correlation between the presence of the *mcr-4.3* gene and colistin resistance (65 mg/L) in the clinical *A. baumannii* 597A isolate.

## 3. *Acinetobacter baumannii* Virulome—Virulence Factors and Mechanisms

In addition to the resistome of MDR, XDR, and PDR *A. baumannii* isolates, this opportunistic pathogen employs a variety of virulence factors and mechanisms facilitating survival, which enhances the colonisation and subsequent infection of susceptible hosts. These virulence factors include but are not limited to; cellular envelope factors, outer membrane proteins, secretion systems, phospholipases, and biofilm formation, which concomitantly contribute to the pathogenicity of *A. baumannii* (extensively reviewed by Harding et al. [16]) (Figure 1). The cellular envelope factors, including glycoconjugates or glycans (carbohydrates) such as capsular polysaccharides (CPS), LPS, glycosylated proteins, and peptidoglycan, provide an interface between *A. baumannii* and its environment, thus facilitating survival and persistence [16] (Figure 1). For example, the CPS of *A. baumannii* has been associated with water retention, which facilitates desiccation tolerance. Tipton et al. [39] observed that acapsular (CPS absent) *A. baumannii* AB5075, exhibited a 2.5-fold decrease in desiccation tolerance in comparison to capsular parental strains. In addition, the acapsular (Δ*wzc*) mutant strain exhibited an 8-log reduction in the colony forming units (CFU) at 24 h post-infection of a murine (mouse lung/lungs) model, in comparison to the wild-type, capsular (*wzc*) strain, indicating that the capsule functions as an important virulence factor in infection and pathogenesis. The LPS has also been linked to desiccation tolerance with Boll et al. [40] demonstrating the association between lipid A acetylation and desiccation tolerance in *A. baumannii*. Moreover, strains of *A. baumannii*, with LPS devoid of the hepta-acylated lipid A, exhibited decreased desiccation tolerance, which was proposed to be due to an increased membrane fluidity resulting in the leakage of water and nutrients [40]. In addition to OMPs, several secretion systems have been detected and associated with virulence amongst *A. baumannii* strains, including the Type II secretion system (T2SS), Type V secretion system (T5SS), and Type VI secretion system (T6SS) [16] (Figure 1). The T2SS facilitates the excretion of toxins, hydrolytic enzymes (lipases, lipoproteins, and proteases), aids in the acquisition of nutrients, and is required for in vivo survival and virulence [41,42]. Johnson et al. [42] then demonstrated the association between the T2SS, secretion of lipase (LipA), and pathogenicity in *A. baumannii* using ∆*gspD* (GspD: outer membrane pore) and ∆*gspE* (GspE: ATPase) mutants. Through the generation of *A. baumannii* ∆*gspD* and ∆*gspE* mutants, decreased LipA secretion was achieved resulting in decreased growth and significantly reduced in vivo fitness (decreased colonisation of spleen and liver) in murine (CBA/J mice) models. Therefore, T2SS was proposed to facilitate nutrient acquisition through the excretion of lipase, which allowed for in vivo colonisation, thus contributing to the pathogenesis of *A. baumannii*.

Other membrane-associated structures, including the OMPs (OmpW, CarO, OprF, OprD, AbuO, TolB, DcaP, Oma87/BamA, NmRmpM, CadF, and LptD), have also been identified as virulence factors in *A. baumannii* [48] (Figure 1). The most abundant and extensively studied *A. baumannii* OMP is OmpA (previously referred to as Omp38), which facilitates cell adherence and invasion (observed to be dependent on the host cell type), with respiratory tract epithelial cells (bronchial (NCI-H292) and laryngeal (HEp-2) cells) being more susceptible to infection in comparison to non-respiratory tract epithelial cells (cervical carcinoma (HeLa) cells) [49].

Through the implementation of numerous surface and extracellular-associated structures, *A. baumannii* is also able to form biofilms, which has been characterised as a major virulence factor contributing to the bacterium’s pathogenicity [6,50]. Biofilm formation has also been associated with increased pathogenicity, with Khalil et al. [51] reporting on an increased killing rate (50 to 90%), observed in a larva (*G. mellonella*) model, by strong biofilm-forming *A. baumannii* strains in comparison to moderate and weak biofilm-forming strains. Structures such as the chaperone-usher (Csu) pili and CsuA/BABCDE-independent short pili system, as well as biofilm associated-proteins (Bap or BAP), allow *A. baumannii* to adhere to both abiotic (polyethylene, polystyrene, titanium, and Teflon) and biotic (human bronchial epithelial (H_292_) cells and neonatal keratinocyte cells) surfaces [52] (Figure 1). Two Bap-like proteins (BLP: BLP1 and BLP2) have also been identified in different *A. baumannii* strains, facilitating both adherence and biofilm formation on bronchial epithelial (A549) cells [53] (Figure 1). Moreover, extracellular polymeric substances form part of *A. baumannii* biofilms and are composed of poly-β-(1-6)-N-acetylglucosamine and extracellular DNA, providing structural support and intracellular connectors, which allows for biofilm formation under diverse environmental conditions [6]. Biofilm formation and maintenance have also been associated with intracellular communication, mediated by 3′,5′-cyclic diguanylic acid and quorum sensing, which is facilitated by AtaI autoinducer synthase and the AbaR cognate receptor [50] (Figure 1). Additionally, the association or correlation between biofilm formation and antibiotic resistance has been extensively investigated [54,55,56]. For example, Thummeepak et al. [54] investigated the association between biofilm formation, antibiotic resistance phenotype, and virulence genes in clinical *A. baumannii* (*n* = 225) isolates, with 86.2% of the strains characterised as MDR, of which 76.9% were biofilm producers. The biofilm formation genes, *ompA* and *bap* were further linked/associated with the MDR phenotype of the *A. baumannii* isolates.

## 4. Biological Control Strategies for MDR, XDR, and PDR *A. baumannii*

Novel control strategies to combat MDR, XDR, and PDR *A. baumannii* are thus urgently required and were mandated by the World Health Organisation during the development of the global priority pathogen list to assist in prioritising the research and development of new and effective antimicrobial treatments [57]. Accordingly, there has been an upsurge in research focusing on biological control strategies to combat bacteria resistance to many commercially available antibiotics, as these approaches are considered environmentally friendly, cost-effective, self-sustaining, highly specific, and result in low resistance frequencies. While biological control strategies, including ribosomally synthesised primary metabolites such as bacteriocins (garvicin KS, nisin, and enterocin-A and -B), have exhibited antimicrobial activity against *A. baumannii* [58,59], the current review will focus on biological control strategies including non-ribosomally synthesised secondary metabolites such as biosurfactants (lipopeptides and glycolipids), predatory bacteria (*B. bacteriovorus*), and bacteriophages.

### 4.1. Biosurfactants

Biosurfactants are non-ribosomally synthesised, surface-active secondary metabolites produced by various actively growing microorganisms including bacteria, yeast, and filamentous fungi [8]. Due to their functional properties (stable under various pH, temperature, and ionic fluctuations; biodegradable; low toxicity; display emulsifying and demulsifying capacity), several companies [Allied Carbon Solutions (Sophorolipids), AGAE Technologies (rhamnolipids), Kaneka Corporation (surfactin) and Toyobo (mannosylerythritol lipids)] produce and apply biosurfactants, approved by the United States Food and Drug Administration (US FDA), in various industries such as pharmaceutical/medical, cosmetic, food, petroleum, wastewater treatment, textile, pesticide, biodegradation, and agricultural industries [60]. Particularly lipopeptide and glycolipids are of extreme interest to the pharmaceutical/medical industry, as various classes exhibit broad-spectrum antimicrobial, antibiofilm, and antiadhesive activity [8,9,10]. The chemotherapeutic potential of these metabolites is primarily attributed to the proposed mode of action of lipopeptides and glycolipids, which target the cellular membrane through a detergent-like and/or flip-flop mechanism (transmembrane lipid translocation) [13].

Therefore, as the mode of action is multimodal, it is hypothesised that the extensive resistome and virulome of A. baumannii would not influence the activity of the biosurfactants, highlighting the potential pharmaceutical and medical application of the secondary metabolites in various in vitro and in vivo applications [13].

#### 4.1.1. Lipopeptide Biosurfactants

##### Surfactin

The antibacterial activity of the lipopeptide surfactin, primarily produced by *Bacillus* spp., has been reported against various Gram-negative bacteria, including MDR and XDR *A. baumannii*. For example, Havenga et al. [61] investigated the susceptibility of MDR and XDR *A. baumannii*, *Pseudomonas aeruginosa* (*P. aeruginosa*), *Escherichia coli* (*E. coli*), and *Klebsiella pneumoniae* (*K. pneumoniae*) strains to a crude surfactin extract (containing C_13–16_ surfactin analogues) produced by *Bacillus amyloliquefaciens* (*B. amyloliquefaciens*) strain ST34. Results indicated that the crude surfactin (C_13–16_ surfactin analogue) extract (10.00 mg/mL) retained antimicrobial activity against all (100%) *A. baumannii*, *P. aeruginosa*, *E. coli,* and *K. pneumoniae* strains classified as MDR, XDR, and colistin resistant. While limited research is available on the antibiofilm and antiadhesive activity of surfactin against specifically MDR, XDR, and PDR *A. baumannii* strains, the antiadhesive and antibiofilm activity against other Gram-negative and Gram-positive bacteria has been demonstrated. For example, Meena et al. [62] observed that purified surfactin (100 µg/mL) obtained from *Bacillus subtilis* (*B. subtilis*) KLP2015 exhibited increased antimicrobial activity against *K. pneumoniae*, *Salmonella enterica* (*S. enterica*), *Staphylococcus aureus* (*S. aureus*), and *E. coli*, with 58.1% and 47.86% antibiofilm activity recorded for *S. aureus* and *Pseudomonas* sp., respectively. The purified surfactin crude extract (100 µg/mL) was further observed to exhibit antitumor activity against five cancer cell lines (HCT-15, Hep2-C, L-132, MCF-7, and NIH/3T3), with increased cytotoxicity recorded against HCT-15 (80.1 ± 1.92%) in comparison to the normal HaCaT (31.45 ± 2.58%) cell line.

In addition to the minimal cytotoxic effects exhibited on healthy cell lines, the toxicology of surfactin has been studied using in vivo models [63,64]. Hwang et al. [64] investigated and compared the antibacterial activity of surfactin C and polymyxin B in a murine (ICR mice and Sprague-Dawley (SD) rats) model infected with *E. coli* O111:B4. Overall, survival rates of 53.3%, 73.3%, and 73.3% were recorded at surfactin C concentrations of 5, 10, and 25 mg/kg, respectively, while survival rates of 86.6% were recorded at 1 mg/kg polymyxin B. Although increased survival rates were recorded for the polymyxin B, the compounds have been shown to exhibit adverse effects (neurotoxicity and nephrotoxicity) at 1 to 2 mg/kg, while research has indicated that surfactin only becomes toxic at significantly higher concentrations [LD_50_ (50% lethal dose)] of 100 mg/kg [63]. Thus, while additional in vitro and in vivo studies are required to confirm the safety and efficacy of surfactin, current results substantiate the potential use of this metabolite for the treatment of MDR, XDR, and PDR *A. baumannii*-associated infections.

##### Serrawettin

The antibacterial activity of the lipopeptide serrawettin, produced by *Serratia* spp., has been reported against various Gram-negative bacteria, including *A. baumannii* [10]. In a study conducted by Clements et al. [10] three *Serratia marcescens* (*S. marcescens*) strains [pigmented 1 (P1); nonpigmented 1 (NP1) and 2 (NP2)] produced crude extracts containing serrawettin W1, serrawettin W2, glucosamine derivative A, and prodigiosin (P1 only). All three crude extracts (P1, NP1, and NP2) exhibited antimicrobial activity (1 mg/mL) against *A. baumannii* ATCC 19606, whereas only the P1 and NP2 crude extracts were effective in inhibiting the clinical XDR *A. baumannii* strain AB 3. In a follow-up study, the antibiofilm and antiadhesive capabilities of the biosurfactants produced by *S. marcescens* P1 and NP1 against *P. aeruginosa* S1 68 and *Enterococcus faecalis* (*E. faecalis*) S1 were demonstrated [65]. At a P1 crude extract concentration of 2.5 and 50 mg/mL and NP1 crude extract concentration of 5 and >50 mg/mL, *P. aeruginosa* S1 68 and *E. faecalis* S1 biofilms (formed on polystyrene) were dislodged and removed, respectively. In addition, an antiadhesive activity of 99.07% and 94.39%, and 95.83% and 93.11%, against *P. aeruginosa* S1 68 and *E. faecalis* S1, respectively, was observed at a P1 and NP1 crude extract concentration of 50 mg/mL.

Shanks et al. [66] investigated the haemolytic and cytotoxic effect of serratamolides (serrawettin homologues) against red blood cells (from C57BL/6 mice) as well as human bronchial (A549) epithelial cells and human corneal limbal epithelial cells (HCLE) monolayers. The serratamolides exhibited haemolytic activity at a concentration of 1 mg/mL and cytotoxicity towards human bronchial (A549) epithelial cells and HCLE monolayers at 50 µg/mL. While the cytotoxicity results may limit the use of this lipopeptide, Clements et al. [10] showed that the crude extracts (containing serrawettin W1, serrawettin W2, glucosamine derivative A, and prodigiosin) produced by the *S. marcescens* P1, NP1, and NP2 strains, exhibited no haemolytic activity at a concentration of 1 mg/mL. Additional in vitro and in vivo studies are thus required to investigate the safety of serrawettin lipopeptides before the compounds can be implemented for the biological control of MDR, XDR, and PDR *A. baumannii*.

#### 4.1.2. Glycolipid Biosurfactants: Rhamnolipids

The antibacterial activity of glycolipids such as rhamnolipids, produced by various bacterial species, has been demonstrated against several antibiotic-resistant Gram-positive and Gram-negative bacteria. For example, Ndlovu et al. [8] demonstrated the broad-spectrum antimicrobial activity of a rhamnolipid crude extract (1 mg/mL), consisting of congeners of mono- and di-rhamnolipids, produced by *P. aeruginosa* ST5, against antibiotic-resistant Gram-positive and Gram-negative bacteria, including *S. aureus* ATCC 25923, methicillin-resistant *S. aureus* (MRSA) *Xen 30*, *K. pneumoniae* (ATCC 10031, P2, P3, k2a), enteropathogenic *E. coli* B170, *S. enterica,* and *Acinetobacter* sp. F1S6. Moreover, while limited research has been published on the antiadhesive and antibiofilm activity of rhamnolipids against specifically MDR, XDR, and PDR *A. baumannii* strains, the antiadhesive, antibiofilm, and cytotoxic activity of rhamnolipids has been demonstrated. Aleksic et al. [67] investigated the antibacterial, antibiofilm, and cytotoxic properties of a di-rhamnolipid (Rha-Rha-C_10_-C_10_, Rha-Rha-C_8_-C_10_, and Rha-Rha-C_10_-C_12_) produced by *Lysinibacillus* sp. BV152.1, which exhibited increased antiadhesive and antibiofilm activity towards *P. aeruginosa* PAO1 (NCTC 10332) biofilms at 50 µg/mL and 75 µg/mL, respectively. Additionally, the authors observed that the rhamnolipid exhibited no cytotoxic activity toward human lung fibroblasts (MRC5) cell lines at a concentration of 100 µg/mL. Thanomsub et al. [68] further reported that rhamnolipids, produced by *P. aeruginosa* B189, exhibited cytotoxic activity against a breast cancer cell line (MCF-7) at a MIC of 6.25 µg/mL, with no toxicity recorded against healthy vero cell lines at the tested concentration range of 0.78 to 50 µg/mL. Tawfeeq and Yesser [69] then provided insight into the in vivo antimicrobial potential of rhamnolipids. Using a murine (*Mus musculus*) model superficially infected with *S. aureus* and *P. aeruginosa*, the authors showed that at a concentration of 30 mg/mL purified PS10 and PS16 rhamnolipid, the infection cleared within 10 to 12 days, in comparison to the untreated mice, where a 17-day recovery period was required. Thus, while the in vivo treatment efficacy of rhamnolipids has been demonstrated, the efficacy of these secondary metabolites against specifically MDR, XDR, and PDR *A. baumannii* strains needs to be investigated and validated.

#### 4.1.3. Biosurfactant Applications: Current Limitations and Potential Mitigation Strategies

While limited research has been published on the resistance of pathogenic Gram-negative bacteria to the various classes of biosurfactants, several studies have been published on mechanisms by which biosurfactant-producing microorganisms may themselves potentially exhibit resistance to this class of secondary metabolites. Research on biosurfactant resistance by the producer strains has almost exclusively been conducted on the lipopeptide surfactin, produced by *B. subtilis*, and has been associated with: (1) RND-like superfamily type efflux system or other proton motive force dependant transporters [YerP (yerP), YcxA (ycxA), and KrsE (krsE)]; (2) biochemical and biophysical membrane alterations ((i) the transient reduction in branched fatty acids; (ii) the emergence of non-branched C_16:00_ and C_18:00_ fatty acids; (iii) gradual replacement of phosphatidylglycerol and phosphatidylethanolamine with the “stress phospholipid” cardiolipins); and (3) additional enzymatic resistance (hydrolase) (Figure 2) [12,13]. Thus, while the self-resistance mechanisms exhibited by the producer strains are diverse and complex, observations made in these mechanistic studies indicate that biosurfactant resistance amongst other bacteria (including *A. baumannii*) may potentially be mediated through an adaptive change in the cell wall or through the production of inactivating enzymes.

Therefore, to circumvent the potential development of resistance to biosurfactants by the target pathogenic Gram-negative and Gram-positive bacteria, combination therapy could be implemented as studies have reported on the synergistic interaction between biosurfactants and antibiotics [71,72]. For example, Sudarmono et al. [72] investigated the potential synergistic combination of a commercial antibiotic (ampicillin) with surfactin (produced by *B. amyloliquefaciens* MD4-12) against *P. aeruginosa* ATCC 27853. The MIC of surfactin against *P. aeruginosa* ATCC 27853 exceeded 1024 μg/mL, whereas a MIC of 256 μg/mL was recorded for ampicillin. In contrast, the combination of 32 to 512 μg/mL surfactin with 64 μg/mL ampicillin resulted in a greater antimicrobial effect being exhibited against the test organism. Similarly, Samadi et al. [71] demonstrated the combination effect of rhamnolipids (produced by *P. aeruginosa* MN1) with oxacillin against seven *S. aureus* isolates (ATCC 33591 and MRSA1–MRSA6). Rhamnolipid and oxacillin MICs ranged from 25 to 50 µg/mL and 50 to 1600 µg/mL, respectively, against the *S. aureus* ATCC 33591 and MRSA1 to MRSA6 isolates. However, in combination experiments with rhamnolipid (6.25–25 µg/mL) and oxacillin (0.1–6.25 µg/mL), synergism was recorded against MRSA1, MRSA4, and *S. aureus* ATCC 33591, whereas partial synergism was recorded against MRSA2, MRSA3, MRSA5, and MRSA6. Overall, these studies demonstrated the extent to which biosurfactants (lipopeptides and glycolipids) could be used to curtail resistance development and increase or broaden the antimicrobial activity, through the potential re-sensitization of the target bacteria to commercially available antibiotics.

While it is evident that biosurfactants (lipopeptides and glycolipids) could be applied to the pharmaceutical/medical industry as monotherapies or in combination with commercial antibiotics, the commercialisation or large-scale (up-scaling) production remains a major limitation as the process is not cost-effective. To date, several strategies have thus been proposed to reduce or mitigate the overall cost and increase biosurfactant yield for large-scale production, including (1) the use of low-cost substrates; (2) improvement of medium composition through statistical optimisation; and (3) the genetic engineering of biosurfactant producing bacteria to enhance biosurfactant production [73]. For example, biosurfactant (lipopeptides and glycolipids) production has primarily been carried out in synthetic mediums such as mineral salt medium (MSM) with a carbon source (glucose, sucrose, fructose, or glycerol); however, the use of inexpensive and renewable substrates, including waste products from various industries (food, petroleum, wastewater treatment, and agricultural) are being investigated to reduce the cost associated with large-scale production (extensively reviewed by Banat et al. [74]).

Growth medium optimisation has also been recommended to improve biosurfactant yield and reduce the cost associated with large-scale production. Two statistical methods, namely, the Plackett–Burman design (PBD) and the response surface methodology (RSM) have been extensively implemented to optimise media components for enhancing surfactin and rhamnolipid production. The PBD identifies the most important variables (chemical composition of media, pH, osmolarity, temperature, oxygenation, and agitation) affecting the response (biosurfactant production) for further downstream optimisation. In comparison, RSM is a follow-up statistical method of the PBD and allows for the modelling and collective analysis of all the important variables, with the objective to optimise the response [75]. Wibisana et al. [76] implemented RSM statistical optimisation to screen the significant factors, including production medium components (carbon and nitrogen source, monosodium glutamate, magnesium sulphate, dipotassium phosphate, potassium chloride, and trace elements) and environmental conditions (pH and temperature), which affected surfactin production by *B. amyloliquefaciens* MD4-12. Through the optimisation process, the authors increased *B. amyloliquefaciens* MD4-12 surfactin production by 2.4-fold (1.25 g/L) in comparison to the un-optimised conditions.

It should, however, be noted that while the implementation of statistical methods has been effective in the optimisation of biosurfactant production, secondary metabolite production may be restricted by the low concentration and productivity of the wild-type bacterial strain [77]. Consequently, research has shifted towards the genetic engineering of bacteria to enhance biosurfactant production. Initial attempts were primarily focused on the generation of random mutations (mutagenesis), which were proposed to result in increased biosurfactant production, through ultra-violet (UV) radiation and *N*-methyl-*N*’-nitro-*N*-nitrosoguanidine (NMG) exposure. For example, UV radiation treatment of *B. subtilis* ATCC 21332 resulted in a 3-fold increase in surfactin production in comparison to the wild-type *B. subtilis* ATCC 21332 strain [78]. Similarly, random mutagenesis with NMG was applied to *Bacillus licheniformis* KGL11, resulting in a 12-fold increase in surfactin production [79]. More recently, genetic engineering approaches, such as recombinant DNA technology, the overexpression of extracellular peptides, substitutions, replacement, and modification of amino acids as well as gene/gene cluster knockouts (extensively reviewed by Jimoh et al. [80]), have been implemented to improve biosurfactant production by specific bacterial strains including *Bacillus* and *Pseudomonas* spp. For example, Jung et al. [81] genetically engineered a *B. subtilis* 1012WT to overexpress extracellular peptides, ComX (*comX*) and PhrC (*phrC*), which are associated with the stimulation of surfactin production under low-cell densities. The recombinant *B. subtilis* pHT43-*comXphrC* strain produced 6.4-fold (135.1 mg/L) more surfactin in comparison to the wild-type (1012WT) strain. Genetic engineering could thus be applied to enhance biosurfactant production; however, biosurfactant yield improvement (optimisation) will only be fully realised once the regularity mechanisms of biosurfactant production are fully elucidated [11].

### 4.2. Predatory Bacteria: Bdellovibrio bacteriovorus

*Bdellovibrio bacteriovorus* is one of the most extensively studied predatory bacteria in the Bdellovibrionaceae family and is characterised by a periplasmic biphasic or dimorphic life cycle. In the attack phase, motile, free-swimming predator cells scavenge or search for prey bacteria. Once the predator attaches to the potential prey, it invades the prey cell forming the bdelloplast (structure in which progeny cells are produced), whereafter it produces various hydrolytic enzymes, which degrade the prey cell components in the growth phase [82]. During the growth phase, bdelloplast formation results in the rounding morphology of the prey cell as a result of peptidoglycan modifications, followed by predator septation and the release of flagellated progeny cells (Figure 3) [82,83,84]. As a result of its obligate predatory lifestyle, *B. bacteriovorus* has been investigated as a biocontrol agent in the aquaculture, agriculture, water, and sanitation industries [15].

Moreover, the application of *B. bacteriovorus* as a live antibiotic in the pharmaceutical and medical industry has gained interest due to the increased prevalence of infections caused by MDR, XDR, and PDR bacteria, including strains of *A. baumannii* [85,86]. Kadouri et al. [85] assessed the predatory efficiency of two *B. bacteriovorus* strains, namely, *B. bacteriovorus* 109J and *B. bacteriovorus* HD100 (ATCC 15356), in co-culture with the MDR Gram-negative pathogens *A. baumannii, E. coli, K. pneumoniae, P. aeruginosa,* and *Pseudomonas putida* (*P. putida*). *Bdellovibrio bacteriovorus* 109J was able to prey on 93% of the host bacterial strains with pronounced predatory activity exhibited towards *A. baumannii* AB276, *K. pneumoniae* AZ1169, and *P. aeruginosa* GB771. Similarly, *B**. bacteriovorus* HD100 was able to prey on 100% of the prey bacteria with significant reductions in cell counts recorded for *A. baumannii* AB276, *E. coli* YD438, *K. pneumoniae* AZ1093, and *P. putida* YA241. Correspondingly, Dharani et al. [86] investigated the predatory efficacy of *B. bacteriovorus* strains (109J and HD100) on planktonic and sessile cultures of colistin sensitive (wild-type) and *mcr-1* positive, colistin-resistant *A. baumannii*, *E. coli*, *K. pneumoniae,* and *P. aeruginosa* strains. Based on the results obtained, colistin sensitive and resistant planktonic cultures of the prey bacteria were susceptible to both *B. bacteriovorus* strains 109J and HD100. Moreover, colistin resistance had no significant influence on the antibiofilm activity of the predators, as *B. bacteriovorus* 109J and HD100 reduced the biofilm density of the wild-type *A. baumannii* by 51% and 50%, respectively, which was comparable to the removal of the *mcr-1* positive, colistin-resistant *A. baumannii* biofilm (61% and 57%, respectively).

#### *Bdellovibrio bacteriovorus* Therapy: Current Limitations and Potential Mitigation Strategies

While *B. bacteriovorus* resistance has not been detected amongst *A. baumannii* strains, previous studies have observed that certain bacterial species exhibit (1) population-based resistance (plastic phenotypic resistance) and (2) cell-wall based resistance (production of procrystalline protein or S-layer), when exposed to various predatory strains (Figure 3) (Shemesh and Jurkevitch 2004). Thus, as *Bdellovibrio* spp. have been observed to exhibit inherent resistance to β-lactam antibiotics and antifolates, it has been hypothesised that *Bdellovibrio* spp. could serve as potentiators or adjuvants to conventional antibiotics (penicillins, carbapenems, and trimethoprim) or antimicrobials, which, in turn, may mitigate the development of predator and/or antibiotic resistance [82,83]. Im et al. [90] demonstrated the antimicrobial efficacy of *B. bacteriovorus* HD100 in combination with violacein (bis-indole pigment derived from *Pseudoduganella violaceinigra* sp. NI28) against Gram-positive and Gram-negative bacteria including MDR *S. aureus*, *Bacillus cereus* (*B. cereus*), *Staphylococcus epidermidis* (*S. epidermidis*), *E. coli*, MDR *K. pneumoniae,* and MDR *A. baumannii* in mono- and dual-culture experiments as well as polymicrobial cultures. Violacein in combination with *B. bacteriovorus* HD100, resulted in a 99.8% reduction in *S. aureus* and *A. baumannii* cell counts, respectively, in the monoculture experiments. In addition, treatment (violacein and *B. bacteriovorus* HD100) of dual cultures of either *A. baumannii* or *K. pneumoniae* with *S. epidermidis*, resulted in an average cell count reduction of 99.3% for each of the pathogens. Moreover, the treatment of a polymicrobial culture consisting of *S. aureus*, *A. baumannii*, *B. cereus,* and *K. pneumoniae*, with violacein and *B. bacteriovorus* HD100, also resulted in a 99.96% cell count reduction. Through the development of a novel liquid assay, Marine et al. [82] proceeded to screen 21 commercial antibiotics in combination with *B. bacteriovorus* HD100. The aim of the study was to determine whether the combination of the commercial antibiotics with the predatory bacterium would produce an increased antimicrobial efficacy and reduce the development of antibiotic resistance. Trimethoprim then exhibited the lowest activity against *B. bacteriovorus* HD100 and highest activity towards *E. coli*, which implied that this antibiotic could potentially be employed in combination therapy with the predatory bacterium. Furthermore, the development of this novel screening method implies that other *B. bacteriovorus* strains could potentially be screened and applied in combination with commercial antibiotics, broadening the antibacterial application of *B. bacteriovorus* and simultaneously circumventing the potential development of predation and antibiotic resistance.

In addition to predatory resistance, environmental conditions, or factors (viscosity, osmolarity, and temperature) have been identified as a potential limitation for the application of the predatory bacteria as biological control agents [83]. For example, in the study conducted by Im et al. [91], serum albumin and osmolarity (environmental factor) were observed to inhibit *B. bacteriovorus* HD100 predation on *E. coli* MG1655, *K. pneumoniae* (clinical isolate), and *S. enterica* KACC 11595 in a human serum model. Using different concentrations of NaCl [0.65 to 1% (*w*/*v*)] to adjust the osmolarity of the HEPES buffer, the authors observed a reduction in the predation activity of *B. bacteriovorus* HD100 at an osmolarity greater than 200 mOsm/kg (milliosmoles per kilogram) (0.65% NaCl), whereas predation was completely inactivated at 250 mOsm/kg (0.82% NaCl). Stabilisation of the predatory bacterium could, however, be carried out through biopolymeric encapsulation [92]. Cao et al. [92] implemented spray drying to encapsulate *Bdellovibrio* sp. strain F16 in gelatin, allowing the predatory bacterium to remain viable at a concentration of 3.5 × 10^7^ plaque-forming units (PFU)/g after 120 days at room temperature. Additionally, the encapsulated *Bdellovibrio* sp. strain F16 retained predatory activity and exhibited antimicrobial activity towards shrimp-pathogenic vibrios at a concentration of 0.8 mg/L. Thus, while the development of bioactive biopolymers for encapsulation of the predator may allow for environmental stabilisation and long-term storage, the implementation of a living bacterium for the treatment of MDR, XDR, and PDR infections may be problematic due to several pharmaceutical and medical legislation and regulations.

The use of *B. bacteriovorus*-derived enzymes, which have been demonstrated to exhibit antimicrobial and antibiofilm activity, may thus be a more feasible endeavour [83,93]. For example, while *A. baumannii* was not investigated, Monnappa et al. [93] demonstrated the antimicrobial and antibiofilm efficacy of extracellular enzymes (serine proteases Bd1962, Bd2269, carboxypeptidase Bd0306, and DNases) produced by a host-independent (HI) *B. bacteriovorus* HD100 strain against the Gram-positive bacterium *S. aureus* KACC 10768. A 10% volume of cell-free HI *B. bacteriovorus* HD100 spent media or supernatant was capable of disrupting > 75% of the *S. aureus* KACC 10768 biofilm within 24 h. In addition, exposure to the HI *B. bacteriovorus* HD100 supernatant, decreased the virulence-associated features (dispersal of biofilm and decreased invasion of human epithelial (MCF-10a) cells) of *S. aureus* KACC 10768 cells, further highlighting the potential biological control application of the hydrolytic enzyme derived from predatory bacteria. Thus, while research on the stability of *B. bacteriovorus*-derived enzymes is limited, enzyme stabilisation could be implemented through immobilization by physical adsorption, ionic and covalent bonds, and various other techniques such as entrapment, encapsulation, and cross linking [94]. However, for the commercial application of *B. bacteriovorus* or *B. bacteriovorus*-derived enzymes, the large-scale production (or up-scaling) and subsequent downstream processing methods need to be developed and optimised.

### 4.3. Bacteriophages

Bacteriophages are bacteria-specific viruses consisting of double-stranded or single-stranded DNA or RNA enclosed within a protein coat. They exhibit either a lytic (virulent) life cycle, where they kill the infected host cells, or a lysogenic (temperate) life cycle, where they integrate into the host genome, or exist as plasmids within the host cell (referred to as a prophage) (Figure 4) [95]. Lytic bacteriophages are of particular interest as biological control agents as they are highly host specific allowing for the precise or targeted elimination of bacteria without exhibiting nontarget bacterial interactions [14]. Accordingly, bacteriophage-based biological control products have been approved by the US FDA, Canadian Environmental Protection Agency, and European Food Safety Agency for use in the agricultural and food industries [96].

In addition, Wintachai et al. [99] observed the antibiofilm activity of a *Siphoviridae* bacteriophage (AB1801) against XDR *A. baumannii.* Results indicated that the AB1801 bacteriophage was capable of inhibiting biofilm formation by 66% (10^8^ plaque-forming units (PFU)/well) and reduced preformed biofilms by 70% (10^8^ PFU/well) following a 24 hour exposure period. Recently, Jiang et al. [101] demonstrated the efficacy of a *Myoviridae* bacteriophage (Abp9) against PDR *A. baumannii* biofilms, with results indicating a 72.2% biofilm reduction within 2 hours.

The therapeutic potential of bacteriophages against *A. baumannii* has also been observed in several in vivo studies [102,103]. For example, Jeon et al. [103] investigated and evaluated the in vivo antibacterial potential of two *Myoviridae* bacteriophages (YMC 13/03/R2096 ABA BP or Βϕ-R2096) in two animal models [larva (*G. mellonella*) and murine (C57BL/6 mice)] infected with carbapenem-resistant *A. baumannii.*

Administration of the Βϕ-R2096 bacteriophage to the larva (*G. mellonella*) model infected with *A. baumannii*, resulted in a 50% increased survival rate within 24 hours (post-infection), whereas the bacteriophage administered to the murine (C57BL/6 mice) model increased survival from 30% (multiplicity of infection; MOI = 0.1) to 100% (MOI = 10) within 12 days. In addition, the administration of Βϕ-R2096 to a murine (C57BL/6 mice) model infected with *A. baumannii* resulted in bacterial clearance within 3 days (post-infection) with no mortality or deleterious side effects observed. Moreover, Hua et al. [102] observed that a *Podoviridae* bacteriophage (SH-Ab15519) exhibited antibacterial activity at a lower titre (or MOI) during the treatment of a murine (BALB/c mice) model infected with carbapenem-resistant *A. baumannii*. Overall, a 90% survival rate was recorded at an MOI of 0.1, 1, and 10, in comparison to the 10% survival rate recorded for the non-bacteriophage-treated control group, with no adverse side effects detected.

While reports are limited, bacteriophage therapies have been implemented for the treatment of human *A. baumannii* infections, in cases where all other therapeutic options were exhausted [104,105]. Recently, Tan et al. [104] administered a single-bacteriophage preparation to an 88-year-old patient suffering from chronic obstructive pulmonary disease (lung disease) and type-2 diabetes, who developed hospital-acquired pneumonia (mechanical ventilation associated) caused by carbapenem-resistant *A. baumannii*. Following the administration (viz., nebulisation) of the single Ab_SZ3 (*Siphoviridae*) bacteriophage preparation for 16 days, clearance of the carbapenem-resistant *A. baumannii* was observed with no reappearance recorded. In addition, during the ongoing coronavirus disease 2019 (COVID-19) pandemic, Wu et al. [105] implemented compassionate bacteriophage therapy for the treatment of patients (*n* = 4) suffering from severe acute respiratory syndrome coronavirus 2 (SARS-CoV-2) complicated with carbapenem-resistant *A. baumannii* lung infections. A single bacteriophage (∅Ab121) was administered to one patient (patient 1); however, following treatment (viz., nebulisation) bacteriophage resistance was observed. Thereafter, a combination of two pre-optimised bacteriophages (∅Ab121 and ∅Ab124) were administered to the four patients through nebulisation (patient 1 to 4) and topical applications (patient 3). Bacteriophage therapy in patients 1 and 2 resulted in infection clearance, recovery, and hospital discharge. However, while biological clearance of MDR *A. baumannii* was observed in patients 3 and 4, both patients died due to respiratory failure, with patient 3 also infected with a carbapenem-resistant *K. pneumoniae* infection.

#### Bacteriophage Therapy: Current Limitations and Potential Mitigations Strategies

While bacteriophage-based therapies hold immense potential as biological control agents; the emergence of bacteriophage resistance mediated via: (1) receptor adaptations (mutations of phenotypical alteration resulting in decreased bacteriophage adsorption); (2) host defence systems (molecular pathways preventing or suppressing phage infections); and (3) phage-derived defence systems (molecular pathways facilitating bacterial competition of host), remains a major obstacle, hampering the effective application of this treatment (Figure 4) (extensively reviewed by Egido et al. [98]). The underlying resistance mechanisms have subsequently been identified and exploited in an approach referred to as “bacteriophage steering”, which involves the “exploitation-specific fitness trade-offs” associated with bacteriophage resistance, including antimicrobial re-sensitisation, reduced virulence, and colonisation defects [106,107]. For example, Altamirano et al. [106] isolated a *Myoviridae* bacteriophage (ɸFG02) and an *Ackermannviridae* bacteriophage (ɸCO01). Following co-culturing with the respective *A. baumannii* host strains, bacteriophage resistance was observed, which correlated with the loss of the CPS. Several exploitable fitness trade-offs were, however, subsequently identified, including reduced *A. baumannii* biofilm formation on polystyrene, reduced virulence in a murine (BALB/c mice) model, re-sensitisation to the human complement system as well as alternative bacteriophages (bacteriophages from different families) and antibiotics (β-lactams and fluoroquinolones). Similarly, Wang et al. [107] observed that a colistin-resistant *A. baumannii* ATCC 17978 exhibited resistance to a *Myoviridae* bacteriophage (Phab24) following co-culturing. Bacteriophage resistance was subsequently attributed to LPS (*lpsBSP*) and capsule polysaccharide (amylovoran: *amsE*) biosynthesis alterations. The bacteriophage-resistant *A. baumannii* isolates then exhibited several exploitable fitness trade-offs including decreased in vivo virulence in a larva (*G. mellonella*) model and increased colistin re-sensitisation.

Bacteriophage–antibiotic combinations have also been highlighted as a promising approach to reduce bacteriophage resistance [106]. The increased antibacterial activity of bacteriophage–antibiotic combinations is based on the phenomenon referred to as bacteriophage–antibiotic synergy or phage–antibiotic synergy (PAS) (extensively reviewed by Segall et al. [108]). Bacteriophage–antibiotic synergy has only recently been demonstrated against MDR *A. baumannii*, with Grygorcewicz et al. [109] investigating a bacteriophage cocktail (five bacteriophages) in combination with 10 antibiotics as an antibiofilm strategy. Overall, the combination of the five bacteriophages with 0.25 mg/mL and 0.5 mg/L trimethoprim–sulfamethoxazole, resulted in the highest biofilm biomass reduction of 94.3% and 98.6%, respectively. Additionally, Grygorcewicz et al. [110] investigated the efficacy of the vB_AbaP_AGC01 bacteriophage (*Autographivirinae*) in combination with several antibiotics (ciprofloxacin, gentamicin, and meropenem) against *A. baumannii* ATCC 19606. A synergistic interaction was observed during the combination of bacteriophage AGC01 (MOI = 10) with meropenem (20 mg/L) and ciprofloxacin (10 mg/L) leading to a 99.99% reduction in *A. baumannii* ATCC 19606 cell counts. The antimicrobial activity of the AHC01–antibiotic combination was further investigated using the in vivo larva (*G. mellonella*) model, with the AHC01–meropenem combination resulting in increased larval survival, with a rate of 35% to 77% recorded in comparison to the bacteriophage only control. The PAS phenomenon further validates the use of bacteriophages as a biological control agent against MDR, XDR, and PDR *A. baumannii*; however, additional studies are required to investigate the mode of action as the PAS may be strain specific.

Resistance development could potentially be circumvented using bacteriophage-derived enzymes as the compounds have been shown to exhibit both in vitro and in vivo antimicrobial activity [111,112]. For example, Kim et al. [112] demonstrated the antimicrobial activity of LysSS, a novel phage endolysin (also termed bacteriophage lysins or enzybiotics), against 16 MDR *A. baumannii* strains. Overall, LysSS exhibited antimicrobial activity at a MIC of 0.063 to 0.25 mg/mL against the MDR *A. baumannii* strains. In addition, LysSS exhibited no in vivo cytotoxic effect on human lung (A549) cells below 250 µg/mL. The administration of 125 µg/mL LysSS also resulted in a 40% survival rate in a murine (BALB/c mice) model infected with *A. baumannii*. In addition, endolysins have been combined with commercially available antibiotics for the in vitro and in vivo treatment of *A. baumannii*-associated infections. Blasco et al. [111] combined the endolysin ElyA1 (25 µg/mL) with colistin, which resulted in a fourfold reduction in the colistin MIC against clinical *A. baumannii* strains (*n* = 25). The results were further confirmed in three in vitro assays using larva (*G. mellonella*), murine (BALB/c mice) skin, and lung infection models.

Although the bacteriophage-derived enzymes have been observed to exhibit in vitro and in vivo antimicrobial activity, additional fundamental research is required to investigate environmental stability (temperature, pH, osmolarity, and UV) as fluctuations may influence the biologic control applications. Enzyme stabilisation could be facilitated through the selection of optimal conditions [enzyme concentration, storage buffer, pH, temperature, and the addition of chemical stabilisers (calcium ions and Poloxamer 188)] [113,114]. Bacteriophage endolysins produced in native form often exhibit poor expression or insolubility; therefore, Jun et al. [113] stabilised the recombinant SAL-1 (bacteriophage endolysin) with calcium ions and Poloxamer 188 (stabilised endolysin referred to as SAL200) and subsequently evaluated the in vitro and in vivo antibacterial activity against various *S. aureus* strains. Overall, SAL200 retained in vitro bactericidal activity against both planktonic (0.13 ± 0.03 µg/mL) and sessile (10 µg/mL) *S. aureus* SA1 cultures. In addition, SAL200 was observed to exhibit broad-spectrum antimicrobial activity against various clinical *S. aureus* isolates (*n* = 425). Moreover, SAL200 retained in vivo activity, as murine (ICR mice) models injected with *S. aureus* SA2 (1 × 10^8^ CFU/mouse) remained viable (0% mortality rate: 0/15) at a concentration of 25 mg/kg.

In addition to chemical stabilisation, molecular engineering has been implemented to stabilise thermo-susceptible endolysins as these enzymes have been observed to exhibit a short-term therapeutic shelf-life expectancy, which may limit the development of potential antimicrobial (enzybiotic) compounds [114]. Therefore, to address the limitations regarding transient long-term stability, Heselpoth et al. [114] implemented a FoldX-driven computational protein engineering to increase the thermostability of the PlyC endolysin (derived from the streptococcal C1 lytic phase) catalytic subunit (PlyCA). Through the implementation of computational engineering and visual inspection, eight-point mutations of the PlyC were identified and predicted to be associated with thermostability. One mutation, PlyC (plyCA) T406R, was shown to experimentally increase the thermal denaturation temperature by ~2.2 °C and kinetic stability by 16-fold in comparison to the wild-type endolysin. While the increase in thermal denaturation was modest, the authors highlighted that a single mutation resulted in a pronounced increase (16-fold) in kinetic stability; therefore, multiple advantageous mutations could be induced to additively stabilise an enzyme for antimicrobial applications.

While several different bacteriophage-based products (EcoShield™, ListShield™, and SalmoFresh™) have been commercialised, the development of an effective, constant, and controllable process for large-scale bacteriophage production needs to be optimised. Upscaling has primarily been limited due to the biological nature of the system (growth conditions of host bacterium and bacteriophage infection) and the diverse range of interactions described between bacteriophages and bacteria (bacteriophage resistance mechanisms) (reviewed by García et al. [115]). Several research groups have, however, successfully implemented bioreactors (batch, continuous, or cellstat fermenters) to up-scale bacteriophage production [116,117]. Sochocka et al. [116] successfully implemented a batch (fix volume of nutrients) bioreactor (5 L) to up-scale the production of a T4 bacteriophage using *E. coli* B strain as a bacterial host. Following co-culturing of the T4 bacteriophage and *E. coli* B strain for 24 hours, a bacteriophage titre of 1.2 × 10^16^ PFU/mL was obtained. Comparable results were obtained in a study conducted by Warner et al. [118], where the M13 bacteriophage was produced at a titre of 5.0 × 10^12^ PFU/mL following batch fermentation with XL1-Blue MRF *E. coli*. While batch fermentation is cost effective, it is limited by the volume of the bioreactor equipment, total operation time, and nutrient availability [115]. Therefore, continuous cultivation (constant influx and efflux of nutrients and waste) has been investigated as an alternative for the upscaling of bacteriophage production. Nabergoj et al. [117] achieved a constant T4 bacteriophage titre of 10^9^ bacteriophages per hour in a 1 L cellstat (two bioreactors connected in series) using the bacterial host strain *E. coli* K-12. Furthermore, continuous cultivation systems (chemstat) prevent bacteriophage resistance from developing as two separate bioreactors are used for the growth of bacteria and bacteriophages [115].

## 5. Conclusions and Future Research

The extensive resistome and virulome of the superbug *A. baumannii*, facilitate its survival, proliferation, and epidemic spread in healthcare facilities worldwide. The implementation of biological control strategies including biosurfactants [lipopeptides (surfactin and serrawettin) and glycolipids (rhamnolipids)], predatory bacteria (*B. bacteriovorus*), and bacteriophages could thus potentially be the “silver bullet” solution to combat antibiotic resistance. However, while the antimicrobial activity of biosurfactants, such as surfactin, serrawettin, and rhamnolipids, has been extensively demonstrated against Gram-negative and Gram-positive bacteria, the antiadhesive and antibiofilm activity against MDR, XDR, and PDR *A. baumannii* strains has not been extensively investigated. In addition, fundamental research into potential biosurfactant resistance development as well as the efficacy and safety of these secondary metabolites needs to be conducted using non-mammalian and mammalian models. Moreover, as biosurfactants have been earmarked for their potential pharmaceutical/medical value, research into media optimisation for implementation in cost-effective large-scale production systems (up-scaling) and the application of genetically enhanced producer strains, is required.

While the predatory bacterium *B. bacteriovorus* has been shown to exhibit antimicrobial and antibiofilm activity towards *A. baumannii* in vitro, future studies should investigate the predator–host interaction and host immune response, which would allow for further validation on the safety, functionality, and stability of this biological control strategy. Current research additionally indicates that *B. bacteriovorus* could be used in combination with commercial antibiotics, which may facilitate the re-sensitisation of bacteria to antibiotics, enhance (or broaden) the activity of available antibiotics, and potentially prevent predation resistance development. Furthermore, using bioinformatical analysis and modern genetic engineering approaches, the genes involved in *B. bacteriovorus*-derived enzyme synthesis could be identified, cloned, expressed, and purified for their potential pharmaceutical/medical application.

Similarly, bacteriophages have been observed to exhibit antimicrobial and antibiofilm activity against *A. baumannii*; however, resistance development is frequently detected. Strategies such as “bacteriophage steering” have thus garnered interest as it allows for antibiotic re-sensitisation and virulence reduction. In addition, bacteriophage–antibiotic combinations have been observed to exhibit PAS which could mitigate or prevent resistance development. However, fundamental research into PAS is required as the phenomenon has been reported to be strain specific, which may limit the safety and functionality of the treatment strategy. Bacteriophage-derived enzymes have also been shown to exhibit antibacterial and antibiofilm activity, and research into the stabilisation of the compounds under various environmental conditions (temperature, pH, osmolarity, and UV) is required. This may allow for the development of antimicrobial therapies that remain active during long-term storage and under unfavourable conditions.

## Figures and Tables

**Figure 1 microorganisms-10-01052-f001:**
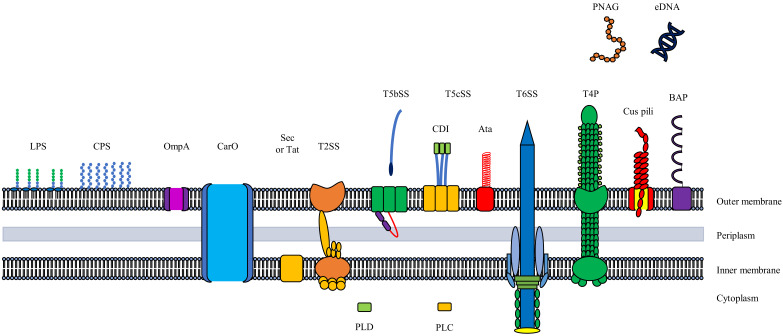
Schematic representation of the virulence factors associated with *A. baumannii* virulome. Cellular Envelope Factors (desiccation resistance, in vivo survival, evasion of host immune response): LPS—Lipopolysaccharide; CPS—Capsular polysaccharide; Outer Membrane Proteins (OMPs) (adherence, invasion, and cytotoxicity): OmpA—Outer membrane protein A; CarO—Carbapenem susceptible porin; SS – Secretions Systems (inter- and intraspecies competition, adherence, nutrient acquisition, in vivo survival): Sec—Secretory pathway; Tat—Twin-arginine system; T2SS—Type II secretion system; T5SS—Type V secretion system; T5bSS—Type Vb secretion system; T5cSS—Type Vc secretion system; CDI—Contact-dependant inhibition; Ata—*Acinetobacter* trimeric autotransporter; T6SS—Type VI secretion system; Phospholipases (invasion, in vivo survival): PLD—Phospholipase D; PLC—Phospholipase C; Twitching and Swarming Motility (in vivo virulence): T4P or TFP—Type IV pili; Biofilm Formation (environmental survival, adherence, and intracellular communication): Cus pili—Chaperone-usher pili; PNAG—Poly-β-(1-6)-N-acetylglucosamine; BAP or Bap—Biofilm-associated protein; eDNA—Extracellular DNA (structures not drawn to scale; adapted from Harding et al. [16]).Two T5SS have been identified amongst *Acinetobacter* spp., namely, Type Vb (T5bSS) and Type Vc (T5cSS). The T5bSS are classified as two-partner secretion (TPS) systems and have been found to be associated with increased adherence to human epithelial alveolar (A549) cells, and in vivo virulence in nematodes (*Caenorhabditis elegans*) and murine (BALB/c mice) models [43]. Another T5bSS, the contact-dependent growth inhibition (CDI) system or CdiA/CdiB system, facilitates bacterial competition through the secretion of the CdiA toxin into the cytoplasm of neighbouring bacteria [44] (Figure 1). Apart from T5bSS, the T5cSS, the *Acinetobacter* trimeric autotransporter (Ata) type Vc secretion system, has been described as multifactorial, facilitating biofilm formation, extracellular matrix/basal membrane protein (collagen IV cell) adhesion, as well as pathogenesis in murine (C57BL/6 mice) models (Figure 1). The Type VI secretion system (T6SS) is primarily associated with bacterial competition (secretions of peptidoglycan hydrolyses and nucleases) and has also been observed to contribute to in vivo virulence in a larva [*Galleria mellonella* (*G. mellonella*)] model [16,45]. This secretion system further facilitates virulence through the release of phospholipases of which phospholipase C (PLC) and phospholipase D (PLD) have been detected and described for *A. baumannii* strains. Three PLDs (PLD1, PLD2, and PLD3) have then been found to mediate human serum resistance (higher propensity to cause bacteraemia), invasion of human bronchial epithelial cells (BEAS-2B), and pathogenicity in murine (C57BL/6 mice) models and larva (*G. mellonella*) models [46] (Figure 1). For example, the authors observed that *pld* triple mutants (Δ*pld1-3* triple mutant) exhibited a reduced (74.2 ± 3.6%) in vivo virulence in comparison to wild-type *A. baumannii* (89.8 ± 2.6%), highlighting the role of phospholipase in the pathogenesis of this opportunistic bacterium. Furthermore, PLC has been observed to contribute to human epithelial cell (FaDu) cytotoxicity, however, only under nutrient-rich conditions and during exposure to chemical stressors such as ethanol [47].

**Figure 2 microorganisms-10-01052-f002:**
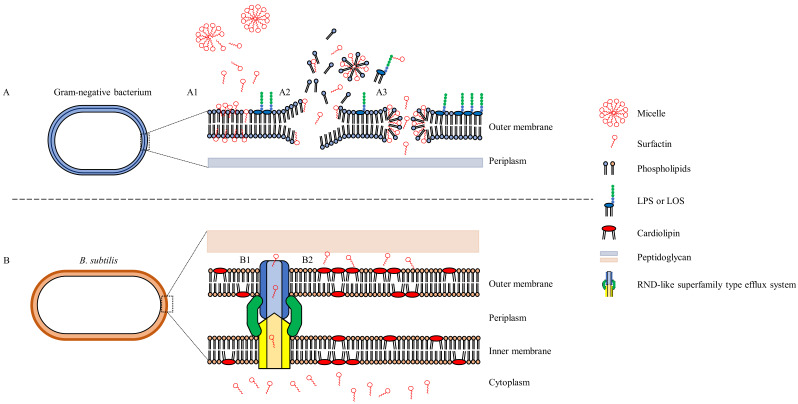
(**A**) Proposed biosurfactant (surfactin) modes of action on Gram-negative bacterium as indicated by (A1) insertion of biosurfactant fatty acid moiety, (A2) membrane disintegration, (A3) pore formation. (**B**) Surfactin resistance mechanisms described amongst *B. subtilis* strains; (B1) RND-like family efflux pump or other PMF dependant transporter and; (B2) Cardiolipin incorporation (adapted from Li et al. [12]; Balleza et al. [70]; Pinkas et al. [13]). LPS—Lipopolysaccharide; LOS—Lipooligosaccharide; RND—Resistance-nodulation-division.

**Figure 3 microorganisms-10-01052-f003:**
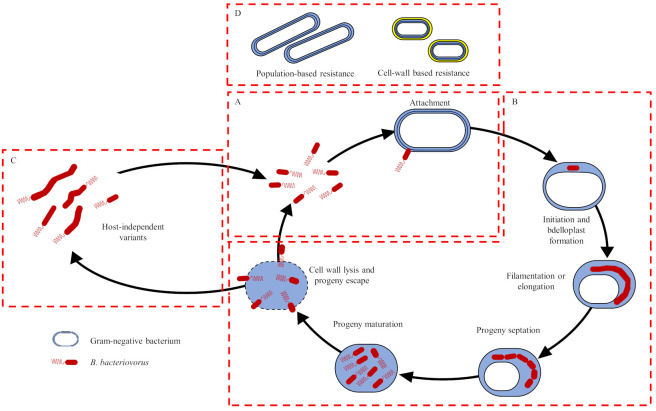
Schematics representation of the proposed life cycle of *B. bacteriovorus* (predator) preying on Gram-negative bacterium (prey): (**A**) Attack phase; (**B**) Periplasmic growth phase; (**C**) Host-independent (HI) phase; (**D**) Proposed predation resistance mechanisms including population-based resistance (plastic phenotypic resistance) and cell wall-based resistance mechanisms (production of procrystalline protein or S-layer) (adapted from Shemesh and Jurkevitch [84]; Marine et al. [82]). The in vivo antimicrobial activity of *B. bacteriovorus* has also been demonstrated, with Shatzkes et al. [87] applying the *B. bacteriovorus* 109J strain to murine (SD rats) models infected with a lethal dose (3.3 × 10^7^ CFU/rat) of *K. pneumoniae* ATCC 43816. The predator strain significantly reduced (99.9%) *K. pneumoniae* ATCC 43816 cell counts in vivo, with no adverse effects observed following treatment. In addition, Findlay et al. [88] demonstrated the efficiency of *B. bacteriovorus* HD100 pre-treatment in a murine model (SKH-1 mice) against a lethal infection (~1000 CFU/100 μL) of *Yersinia pestis* (*Y. pestis*) CO92. The *Y. pestis* CO92 cell counts were significantly reduced (<10 CFU) following administration to the *B. bacteriovorus* HD100 pre-treated murine (SKH-1 mice) models. Additionally, to date, deleterious effects, following the application of *B. bacteriovorus* by ingestion and/or injection, have not been reported, which further validates the application of the predatory bacteria as a potential biological control strategy for the treatment of MDR, XDR, or PDR *A. baumannii* strains [89].

**Figure 4 microorganisms-10-01052-f004:**
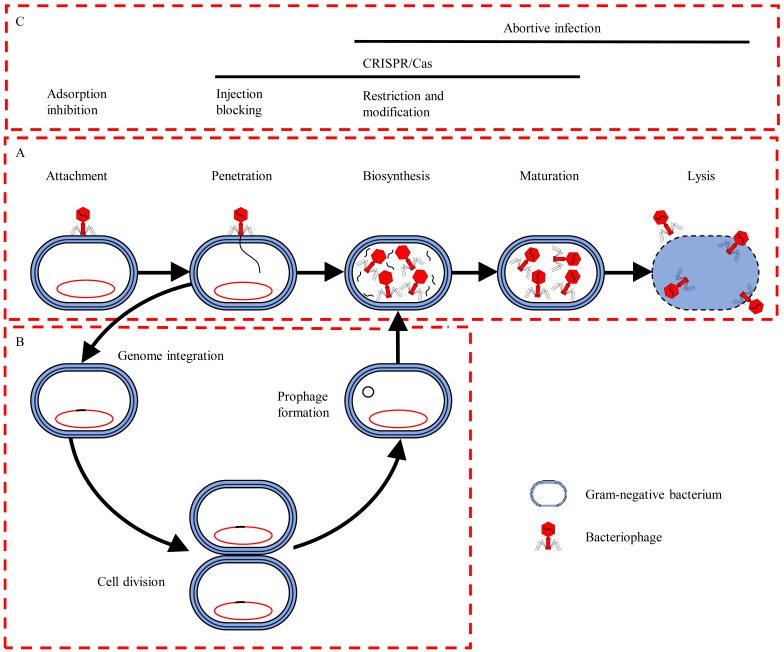
Schematics representation of the (**A**) lytic and (**B**) lysogenic bacteriophage life cycles and (**C**) bacteriophage resistance mechanisms, including receptor adaptations (adsorption inhibitions and injection blocking), host defence systems [clustered regularly interspaced short palindromic repeats (CRISPR/Cas)] and phage-derived defence systems (restriction and modification) (adapted from Hyman and Abedo [97]; Egido et al. [98]). The pharmaceutical and medical application of bacteriophages has also garnered renewed interest globally, due to the rise in the frequency of antibiotic-resistant bacterial infections and the limited availability of new antimicrobial compounds [99]. For example, Merabishvili et al. [42] isolated two bacteriophages, namely, vB_AbaM_Acibel004 (*Myoviridae*) and vB_AbaP_Acibel007 (*Podoviridae*), with Acibel004 exhibiting antibacterial activity towards 75% (*n* = 21/28) of the *A. baumannii* test isolates, while Acibel007 exhibited antibacterial activity towards 60.7% (*n* = 17/28) of the isolates analysed. Correspondingly, Asif et al. [100] isolated a *Myoviridae* bacteriophage (TAC1), which shared high genetic similarity to the *Myoviridae* bacteriophage Acibel004 [4] and exhibited antibacterial activity against 66% (*n* = 21/32) of the MDR *A. baumannii* strains tested.

**Table 1 microorganisms-10-01052-t001:** Antibiotic resistance mechanisms associated with *A. baumannii* (adapted from Lee et al. [17]).

Antibiotic (s)	Resistance Mechanism (s)			Location (s)		Example (s)
β-lactams	Enzymatic		Ambler Class A	C and P		CTX-M-1, CTX-M-2, CTX-M-5, CTX-M-8, CTX-M-9, CTX-M-15, and CTX-M-43
				C and P		CARB-4 and CARB-10
				P		GES-1, GES-5, GES-11, and GES-14
				P		KPC-2, KPC-3, KPC-5, and KPC-10
				C and P		PER-1, PER-2, PER-3, PER-7, and PER-8
				P		SCO-1
				P		SPM-1
				C		SHV-5, SHV-12, and SHV-14
				P		TEM-1, TEM-92, and TEM-116
				C, P and I		VEB-1, VEB-3 and VEB-7
			Ambler Class B or metallo-β-lactamase	I		IMP-1, IMP-2, IMP-4, IMP-5, IMP-6, IMP-8, IMP-11, IMP-14; IMP-19, and IMP-55
				C and P		NDM-1, NDM-2, and NDM-3
				I		SIM-1
				I		VIM-1, VIM-2, VIM-3, VIM-4, and VIM-11
			Ambler Class C	C		AmpC
				P		ADC-1–ADC-81
			Ambler Class D	C and P		OXA-23 subtype: OXA-23, OXA-27, OXA-49, OXA-73, OXA-102, OXA-103, OXA-105, OXA-133, OXA-134, OXA-146, OXA-165, OXA-171, OXA-225, and OXA-239
				C and P		OXA-24/40 subtype: OXA-25, OXA-26, OXA-27, OXA-40, OXA-72, OXA-143, OXA-160, OXA-182, and OXA-207
				C and P		OXA-51 subtype: OXA-51, OXA-64–OXA-71, OXA-75–OXA-80, OXA-82–OXA-84, OXA-86–OXA-95, OXA-98–OXA-100, OXA-104, OXA-106–OXA-113, OXA-115–OXA-117, OXA-120–OXA-128, OXA-130–OXA-132, OXA-138, OXA -144, OXA-148–OXA-150, OXA-172–OXA-180, OXA-194–OXA-197, OXA-200–OXA-203, OXA-206, OXA-208, OXA-216, OXA-217, OXA-219, OXA-223, OXA-241, OXA-242, OXA-248–OXA-250, and OXA-254
				C and P		OXA-58 subtype: OXA-58, OXA-96, OXA-97, and OXA-164
				C and P		OXA-143 subtype: OXA-143, OXA-182, OXA-231, OXA-253, and OXA-255
**Antibiotic (s)**	**Resistance Mechanism (s)**			**Location (s)**	**Example (s)**	
β-lactams	Permeability defects	OMP		C	CarO	
				C	OmpA, Omp33, OmpB, Omp25, OmpC, OmpD, and OmpW	
	Efflux pumps	RND		P	AdeABC	
	Target mutation	PBP		C	PBP6b (dacD)	
Aminoglycosides	Enzymatic	AME		C, P, and I	AAC: *aac(6′)-Ib*’*, aac(3)-IIa, aac(3′)-Ia,* and *aac(3′)IIa*	
					ANT: *ant(3′*’*)-IIa, ant(2′*’*)-Ia, ant(2′)-Ia,* and *ant(3′*’*)-IIa;*	
					APH: *aph(3′)-VI, aph(3′)-Via, aph(3′*’*)-Ib, aph(6)-Id,* and *aph(3′)-VIa*	
	Target mutation	RMTases		P	*armA, rmtA, rmtB, rmtC,* and *rmtD*	
	Efflux pumps	RND		P	AdeABC	
		MATE		C	AdeM	
Quinolones	Target mutation	DNA gyrase		C	GyrA	
		DNA topoisomerase		C	ParC	
	Efflux pumps	RND		P	AdeABC and AdeIJK	
		MATE		C	AbeM	
Tetracyclines and Glycylines	Efflux pumps	RND		C and P	AdeABC, AdeIJK, and AcrAB-TolC	
		MFS		C	TetA and TetB	
	Ribosomal protection	Ribosomal dissociation		P	Tet(O) and Tet(M)	
Polymyxins	Target mutation	Lipid A modification		C	PmrA, PmrB, and PmrC	
				P	*mcr-1* and *mcr-4.3*	
		Lipid A loss		P	LpxA, LpxC, and LpxD	
		Membrane stability		C	LpsB, LptD, and VacJ	
		Biotin synthesis		C	LpsB	

AAC—Acetyltransferases; ADC—*Acinetobacter*-derived cephalosporinases; AME—Aminoglycoside-modifying enzymes; ANT—Nucleotidyltransferases; APH—phosphotransferases; C—Chromosome (Chromosomal); CARB—Carbenicillin-hydrolysing β-lactamases; CTX-M—Cefotaximase-Munich; GES—Guiana extended spectrum; I—Integron; IMP—Imipenem metallo-β-lactamase; KPC—*Klebsiella pneumoniae* carbapenemase; MATE—Multiple antibiotic and toxin extrusion; MFS—major facilitator super family; NDM—New Delhi metallo-β-lactamase; OMP—Outer membrane protein; OXA—Oxacillinase; P—Plasmid; PBP—Penicillin-binding protein; PER—*Pseudomonas* extended resistance; RMTases—16S RNA methylase; RND—Resistance-nodulation-division; SCO—Novel class A β-lactamase; SHV—Sulfhydryl variant; SIM—Seoul imipenem metallo-β-lactamase; SPM—São Paulo metallo-β-lactamase; TEM—Temoniera; VEB—Vietnam extended spectrum β-lactamase; VIM—Verona integrin-encoded metallo-β-lactamase.

## Data Availability

Not applicable.

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
