# Peer review of "Biological Control of Acinetobacter baumannii: In Vitro and In Vivo Activity, Limitations, and Combination Therapies"

_microorganisms, 2022, doi:10.3390/microorganisms10051052_

Round 1
Reviewer 1 Report
The authors provide a comprehensive overview on antibiotic resistance mechanisms, virulence factors/mechanisms and potential biological control strategies, especially against antibiotic resistant Acinetobacter baumannii. In my opinion, this is an interesting review, since up till now little is known on “new”, unconventional strategies like Bdellovibrio bacteriovorus or bacteriophage therapies. The authors highlight and critically discuss the possibilities and limitations in a clearly structured form and in an easy-to-understand way. The tables and figures contribute to clarity and good comprehensibility. It is also helpful that the review refers to the reviews of other authors (“extensively reviewed by…” instead of describing their findings again, which would make the manuscript unnecessarily long). I have only a few comments of minor nature:
- 81: It might be helpful to briefly describe the term “antimicrobial categories” (only with few words).
- 100 ff: Can information on the approximate resistance rates be given (at least for the most commonly used)?
Section 3. Virulome – Virulence Factors and Mechanisms: The factors and systems are described, but it would also be interesting to what extent this affects the actual pathogenicity (e.g., manifestation/symptoms?).
What about bacteriocins? Are there any related bacterial species/substances known which are able to inhibit the growth of A. baumannii? If so, please add the information.
Author Response
Please find comments addressing revision recommendations for the article microorganisms-1712888, “Biological control of Acinetobacter baumannii: In vitro and in vivo activity, limitations, and combination therapies”, outlined below. Please note that recommendations by the reviewer will be listed first (bold) followed by the authors response.
Comment 1: The authors provide a comprehensive overview on antibiotic resistance mechanisms, virulence factors/mechanisms and potential biological control strategies, especially against antibiotic resistant Acinetobacter baumannii. In my opinion, this is an interesting review, since up till now little is known on “new”, unconventional strategies like Bdellovibrio bacteriovorus or bacteriophage therapies. The authors highlight and critically discuss the possibilities and limitations in a clearly structured form and in an easy-to-understand way. The tables and figures contribute to clarity and good comprehensibility. It is also helpful that the review refers to the reviews of other authors (“extensively reviewed by…” instead of describing their findings again, which would make the manuscript unnecessarily long). I have only a few comments of minor nature:
Thank you for the comments.
Comment 2: Line 81: It might be helpful to briefly describe the term “antimicrobial categories” (only with few words).
We have added information to clarify or briefly describe the term “antimicrobial categories” in section 2 as follows:
Lines 92 to 100: “Multidrug resistant (MDR) A. baumannii strains are classified as non-susceptible to at least one agent in three or more antimicrobial classes [antipseudomonal carbapenems, antipseudomonal penicillins + beta-(β)-lactamase inhibitors, penicillins + β-lactamase inhibitors, aminoglycosides, antipseudomonal fluoroquinolones, extended-spectrum cephalosporins, folate pathway inhibitors, tetracyclines and polymyxins]; extensively drug resistant (XDR) strains are classified as non-susceptible to at least one agent in all but two or fewer antimicrobial categories (inhibitors of cell wall synthesis, protein synthesis, DNA or RNA synthesis), while pandrug resistant (PDR) A. baumannii strains are classified as non-susceptible to any agent in all antimicrobial categories (Magiorakos et al. 2012).”
Comment 3: Line 100 ff: Can information on the approximate resistance rates be given (at least for the most commonly used)?
Thank you for the comment. We have added information regarding the reported resistance rates of commonly used antibiotics as follows:
Lines 108 to 111: “Carbapenem resistance (imipenem, meropenem and doripenem) has also been increasing, with a resistance rate ranging from 54.7 to 64.0% recorded amongst A. baumannii strains [World Health Organisation Global Antimicrobial Resistance and Use Surveillance System (GLASS) 2021].”
Lines 123 to 125: “Accordingly, A. baumannii exhibits resistance rates ranging from 80.0 to 90.0% against tobramycin, amikacin, and gentamicin (GERM-SA, 2019).”
Lines 133 to 138: “Tetracycline (doxycycline and minocycline) and glycylcycline (tigecycline) resistance (ranging from 0 to 61.7%) has also been associated with two efflux pump systems including, the RND superfamily type efflux systems (AdeABC and AdeIJK), Major Facilitator Superfamily (MFS: TetA and TetB), and ribosomal protection proteins [Tet(M), Tet(W), Tet(O) and Tet(S)] (Beheshti et al. 2020; Foong et al. 2020) (Table 1).”
Lines 139 to 142: “Apart from the efflux systems, quinolone (ciprofloxacin and levofloxacin) resistance, ranging from 75.0 to 97.7%, has been found to be associated with mutations of the DNA gyrase (gyrA and gyrB) and topoisomerase IV (parC), and plasmid mediated quinolone resistance genes (qnrA, qnrB and qnrS) (Butler et al. 2019; Kyriakidis et al. 2021) (Table 1).”
Comment 3: Section 3. Virulome – Virulence Factors and Mechanisms: The factors and systems are described, but it would also be interesting to what extent this affects the actual pathogenicity (e.g., manifestation/symptoms?).
Information relevant to the association or correlation between A. baumannii’s virulence factors and pathogenicity has been added to section 3 as follows:
Lines 181 to 184: “These virulence factors include but are not limited to; cellular envelope factors, outer membrane proteins, secretion systems, phospholipases, and biofilm formation, which concomitantly contributes to the pathogenicity of A. baumannii [extensively reviewed by Harding et al. (2018)] (Figure 1).”
Lines 191 to 194: “In addition, the acapsular (Δwzc) mutant strain exhibited an 8-log reduction in the colony forming units (CFU) at 24 hours post-infection of a murine (mouse lung/lungs) model, in comparison to the wild-type, capsular (wzc) strain, indicating that the capsule functions as an important virulence factor in infection and pathogenesis.”
Lines 223 to 231: “Johnson et al. (2015) then demonstrated the association between the T2SS, secretion of lipase (LipA), and pathogenicity in A. baumannii using ∆gspD (GspD: outer membrane pore) and ∆gspE (GspE: ATPase) mutants. Through the generation of A. baumannii ∆gspD and ∆gspE mutants, decreased LipA secretion was achieved resulting in decreased growth and significantly reduced in vivo fitness (decreased colonisation of spleen and liver) in murine (CBA/J mice) models. Therefore, T2SS was proposed to facilitate nutrient acquisition through the excretion of lipase which allowed for in vivo colonisation, thus contributing to the pathogenesis of A. baumannii.”
Lines 251 to 254: “For example, the authors observed that pld triple mutants (Δpld1-3 triple mutant) exhibited a reduced (74.2 ± 3.6%) in vivo virulence in comparison to wild-type A. baumannii (89.8 ± 2.6%), highlighting the role of phospholipase in the pathogenesis of this opportunistic bacterium.”
Lines 261 to 264: “Biofilm formation has also been associated with increased pathogenicity, with Khalil et al. (2021) reporting on an increased killing rate (50 to 90%), observed in a larva (G. mellonella) model, by strong biofilm forming A. baumannii strains in comparison to moderate and weak biofilm forming strains.”
Comment 4: What about bacteriocins? Are there any related bacterial species/substances known which are able to inhibit the growth of A. baumannii? If so, please add the information.
Section 4. was amended to include information on bacteriocins as outlined below:
Lines 293 to 299: “While biological control strategies including ribosomally synthesised primary metabolites such as bacteriocins (garvicin KS, nisin, enterocin-A and -B), have exhibited antimicrobial activity against A. baumannii (Ankaiah et al. 2018; Chi and Holo 2018), the current review will focus on biological control strategies including non-ribosomally synthesised secondary metabolites such as biosurfactants (lipopeptides and glycolipids), predatory bacteria (B. bacteriovorus) and bacteriophages.”

Reviewer 2 Report
Dears authors
The current review will provide a brief overview of the antibiotic resistance and virulence mechanisms associated with A. baumannii's "persist and resist strategy". The potential of biological control agents, including secondary metabolites such as biosurfactants [lipopeptides (surfact and serrawettin) and glycolipids (rhamnolipids)], as well as predatory bacteria (Bdellovibrio bacteriovorus) and bacteriophages to directly target A. baumannii is discussed in terms. of their activity in vitro and in vivo. Limitations and corresponding mitigation strategies will also be outlined, including the reduction of resistance development by combination therapies, product stabilization and up-scaling manufacturing.
1 - Introduction: the contents and the writing of the general part must be reformed to review the syntax of the theme
3- Discussion: considering the problem of resistance to antibiotics against multidrug-resistant strains of A. baumanii, to deepen all the problems concerning resistance. and correlations with virulence and biofilm factors. Find out more about this by using and citing the following references:
PMID: 34835509 ; PMID: 35402433 ; PMID: 35203791 ; PMID: 35215067 .
3 - Check the bibliographic entries throughout the text, some of which are non-compliant, review some entries in the bibliographic references .
4 - Review the English grammar and in particular the applied scientific English: in particular the verb tenses and the syntax in the discussion.
Author Response
Please find comments addressing revision recommendations for the article microorganisms-1712888, “Biological control of Acinetobacter baumannii: In vitro and in vivo activity, limitations, and combination therapies”, outlined below. Please note that recommendations by the reviewer will be listed first (bold) followed by the authors response.
Comment 1: Dears authors
The current review will provide a brief overview of the antibiotic resistance and virulence mechanisms associated with A. baumannii's "persist and resist strategy". The potential of biological control agents, including secondary metabolites such as biosurfactants [lipopeptides (surfactin and serrawettin) and glycolipids (rhamnolipids)], as well as predatory bacteria (Bdellovibrio bacteriovorus) and bacteriophages to directly target A. baumannii is discussed in terms. of their activity in vitro and in vivo. Limitations and corresponding mitigation strategies will also be outlined, including the reduction of resistance development by combination therapies, product stabilization and up-scaling manufacturing.
Thank you for the comments.
Comment 2: Introduction: the contents and the writing of the general part must be reformed to review the syntax of the theme
Thank you for the comment. The introduction has been reformatted according to the syntax of the theme.
Lines 30 to 86: “Acinetobacter baumannii (A. baumannii) is one of the primary microorganisms linked to hospital-acquired infections such as central line associated bacteraemia, ventilator-associated pneumonia (VAP) as well as meningitis, bioprosthetic tricuspid valve endocarditis, and urinary tract infections (UTIs) (Gordon and Wareham 2010). The global estimated incidence rate of A. baumannii infections is approximately 1 million cases annually, with crude mortality rates ranging from 20 to 80% (Piperaki et al. 2019; Ma et al. 2021). Previous studies have indicated that several risk factors predispose patients to A. baumannii infection including age (premature babies), immunosuppression, prior hospitalisation [exposure to intensive care unit (ICU)], hospitalisation duration, surgery (invasive procedures), presence of medical indwelling devices (intravascular catheters, urinary catheter, or drainage tubes) and prior or inappropriate antimicrobial therapy (Merabishvili et al. 2014).
Moreover, the extensive resistome of A. baumannii hampers the efficacy of mono-therapeutic options and while antibiotic combination therapies have been shown to exhibit in vitro and in vivo activity against various antibiotic resistant strains, clinical trials have not provided sufficient data to confirm that combination therapies are superior for the treatment of A. baumannii infections (Savoldi et al. 2021). In addition, A. baumannii’s virulome, including cellular envelope factors, outer membrane proteins, secretion systems, phospholipases as well as biofilm formation, allows it to persist under unfavourable environmental conditions for extended time periods, enhancing the colonisation and subsequent infection of susceptible hosts (Mea et al. 2021).
There is thus an urgent need for the research and development of alternative or novel approaches which could be used for the treatment of A. baumannii associated infections, with biological control therapeutic options, defined as the elimination or eradication of a population of microorganisms through the introduction of an antagonistic (predatory) microorganism or its associated secondary metabolites, garnering increased interest (Drakontis and Amin 2020). For example, microbially derived secondary metabolites such as biosurfactants have been described as alternative or novel antimicrobials due to functional properties. Lipopeptides and glycolipids are of extreme interest to the pharmaceutical and medical industry as various classes exhibit broad-spectrum in vitro and in vivo antimicrobial, antibiofilm, antiadhesive activity and low cytotoxicity (Ndlovu et al. 2017; Clements et al. 2019a, 2019b). While the biosurfactants exhibit promising functional properties as biological control agents against A. baumannii, the application thereof remains limited due to the potential development of resistance and the high cost associated with commercialisation or large-scale (up-scaling) production (Banat et al. 2010; Li et al. 2015; Pinkas et al. 2020). In addition, biological control agents including predatory bacteria [Bdellovibrio bacteriovorus (B. bacteriovorus)] and bacteriophages, have been investigated as alternative or novel antimicrobials as these approaches are considered self-sustaining, highly specific and results in low resistance frequencies, highlighting their potential use against A. baumannii (Furfaro et al. 2018; Waso et al. 2021). However, while the biological control agents have been observed to exhibit in vitro and in vivo antimicrobial, and antibiofilm activity with limited cytotoxicity (or deleterious effects) following treatment, the potential of developing resistance and environmental stability, are major limitations impeding their potential application against A. baumannii.
The current review will thus provide a brief overview of A. baumannii’s environmental persistence (bacterial survival under unfavourable environmental conditions), and antibiotic resistance strategies, primarily facilitated through virulence factors and antibiotic resistance mechanisms (Harding et al. 2018). In addition, the therapeutic potential of microbial secondary metabolites [biosurfactants (lipopeptides and glycolipids)] as well as biological control agents including predatory bacteria (B. bacteriovorus) and bacteriophages will be discussed in terms of their in vitro and in vivo activity, limitations such as the potential development of resistance, product stabilisation and large-scale (up-scaling) production. Correspondingly, potential mitigations strategies will focus on the methods to curtail resistance development during treatment (combination therapy with commercial antibiotics), product (B. bacteriovorus and bacteriophage derived enzymes) stabilisation for application in the medical/pharmaceutical industries, and large-scale production and optimisation of the biological control agents or their derived products”
Comment 2: Discussion: considering the problem of resistance to antibiotics against multidrug-resistant strains of A. baumannii, to deepen all the problems concerning resistance and correlations with virulence and biofilm factors. Find out more about this by using and citing the following references:
PMID: 34835509; PMID: 35402433; PMID: 35203791; PMID: 35215067.
We agree with the comment regarding the association between antibiotic resistance, virulence and biofilm formation, and information regarding the correlation was added to review as outlined below:
Lines 278 to 284: “Additionally, the association or correlation between biofilm formation and antibiotic resistance has been extensively investigated (Thummeepak et al. 2016; Donadu et al. 2021; Roy et al. 2022). For example, Thummeepak et al. (2016) investigated the association between biofilm formation, antibiotic resistance phenotype and virulence genes in clinical A. baumannii (n = 225) isolates, with 86.2% of the strains characterised as MDR, of which 76.9% were biofilm producers. The biofilm formation genes, ompA and bap were further linked/associated with the MDR phenotype of the A. baumannii isolates.
Comment 3: Check the bibliographic entries throughout the text, some of which are non-compliant, review some entries in the bibliographic references.
The manuscript has been checked and all references are cited in text and in the reference list and vice versa.
Comment 4: Review the English grammar and in particular the applied scientific English: in particular the verb tenses and the syntax in the discussion.
Thank you for the comment. The manuscript has been checked for spelling and grammar.
